# Real-Time Nondestructive Viscosity Measurement of Soft Tissue Based on Viscoelastic Response Optical Coherence Elastography

**DOI:** 10.3390/ma16176019

**Published:** 2023-09-01

**Authors:** Zhixin Liu, Weidong Liu, Qi Chen, Yongzheng Hu, Yurun Li, Xiaoya Zheng, Dian Fang, Hai Liu, Cuiru Sun

**Affiliations:** 1China Automotive Technology and Research Center, Tianjin 300300, China; liuzhixin@catarc.ac.cn (Z.L.); liuweidong@catarc.ac.cn (W.L.); 2Department of Mechanics, School of Mechanical Engineering, Tianjin University, Tianjin 300350, China; qichen_086@tju.edu.cn (Q.C.); hu566788@tju.edu.cn (Y.H.); 2022201026@tju.edu.cn (Y.L.); zhengxiaoya202109@163.com (X.Z.); fangdian@tju.edu.cn (D.F.); 3Tianjin Key Laboratory of Power Transmission and Safety Technology for New Energy Vehicles, Tianjin 300130, China; liuhai@hebut.edu.cn

**Keywords:** acoustic radiation force, optical coherence elastography, viscosity, relaxation time constant, finite element method

## Abstract

Viscoelasticity of the soft tissue is an important mechanical factor for disease diagnosis, biomaterials testing and fabrication. Here, we present a real-time and high-resolution viscoelastic response-optical coherence elastography (VisR-OCE) method based on acoustic radiation force (ARF) excitation and optical coherence tomography (OCT) imaging. The relationship between displacements induced by two sequential ARF loading—unloading and the relaxation time constant of the soft tissue—is established for the Kelvin-Voigt material. Through numerical simulation, the optimal experimental parameters are determined, and the influences of material parameters are evaluated. Virtual experimental results show that there is less than 4% fluctuation in the relaxation time constant values obtained when various Young’s modulus and Poisson’s ratios were given for simulation. The accuracy of the VisR-OCE method was validated by comparing with the tensile test. The relaxation time constant of phantoms measured by VisR-OCE differs from the tensile test result by about 3%. The proposed VisR-OCE method may provide an effective tool for quick and nondestructive viscosity testing of biological tissues.

## 1. Introduction

Biological tissue is mostly viscoelastic. Quantitative measurement of tissue viscosity is important not only for the diagnosis and treatment of diseases but also for the manufacture of tissue mimicking materials [1]. It has been shown that the Kelvin-Voigt model is a good fit for the viscoelastic behavior of the living tissue [2]. Traditional tensile or compressive creep and stress relaxation measurements require destructive cutting of the tissue to prepare test specimens [3]. Clancy et al. [4] proposed the use of the indentation method to assess the viscoelasticity of the skin in vivo, but it is time-consuming and has low accuracy.

Elastography technique based on optical coherence tomography (OCT) has been proposed, which is termed as optical coherence elastography (OCE). Gubarkova et al. [5] proposed a compression OCE (C-OCE) for characterizing linear models and distinguishing between benign and malignant breast lesions. They then used the method to evaluate the nonlinear properties of breast tissue and the Young’s modulus to improve the accuracy of tumor diagnosis [6]. Zykov et al. [7] proposed an OCE method to detect vascular networks. These methods are static—prone methods, that is, they are more suitable for measuring static elastic properties of the tissue. Shah et al. [8] proposed to combine OCT with vibrational analysis of the deformation generated by an acoustic loudspeaker to measure the natural frequency and the tissue modulus. Recently, OCE based on mechanical wave analysis were studied. Acoustic radiation force (ARF) was demonstrated to be effective on generating local displacement within the tissue to excite mechanical waves [9,10]. ARF-OCE can track submicron displacement of the tissue by phase analysis of the interference signals [11,12]. If the tissue is assumed linear elastic, the shear modulus can be obtained from the simple relationship between the propagation speed of the shear or Rayleigh waves and the shear modulus [13,14].

To characterize the viscoelasticity of biomaterials by OCE, the phase velocity dispersion of elastic waves was usually measured then fitted by viscoelastic wave models. So far, almost all the material measured were assumed as Kelvin-Voigt viscoelastic models in OCE. Han et al. proposed to use a Rayleigh wave model (RWM) [15] and a modified Rayleigh-Lamb frequency equation model (mRLFE) [16] to measure the viscoelasticity of soft tissue excited by air pulses. Wang et al. [17] utilized ARF excitation to generate shear waves in biphasic polymer materials and mammalian liver tissue samples. They estimated the complex modulus of the specimens by fitting the shear-wave dispersion. Similar to ARF excitation, Zhou et al. [18] used a customized ultrasound transducer to excite surface acoustic waves. The phase velocity dispersion curve was extracted using a 2D Fourier transform-based phase velocity analysis algorithm. The shear elasticity and shear viscosity were obtained after fitting the dispersion curve into Rayleigh wave dispersion equation. These methods can mostly obtain reasonable elastic modulus. The accuracy of the viscosity has not been well evaluated. The influence of wave reflections has not been well studied; thus, these elastic wave models are only suitable for the viscoelastic quantification of structures with specific boundary conditions. In addition to obtaining the phase velocity dispersion of the elastic waves, the OCT M-B mode imaging scheme was usually applied, which acquires a big amount of data. It is time-consuming for both data acquisition and processing. Elastic-wave-dispersion-based elastography methods need inversion technology based on elastic wave models to quantitatively calculate the viscoelasticity of local tissue. The experimental control, application of inversion theory and data processing are often complicated [19,20], which can easily cause errors in viscosity measurement. 

Methods of steady-state excitation and recovery [21] solve the viscoelastic properties by fitting the experimental displacement to a viscoelastic model. Wijesinghe et al. [22] proposed to measure viscoelastic creep deformation induced in tissue using step-like compressive loading. The change in local strain of the sample was measured between pairs of co-located, complex OCT B-scans. Then, the local strain rate as a function of time was calculated, based on which the viscoelastic strain and time constant of the Kelvin-Voigt model were estimated. This quasi-static imaging method can provide a two-dimensional map of the elastic distribution of tissue but cannot assess diffuse lesions [23]. Additionally, the gravitational loading system is bulky, which is not easy to be adapted to in vivo imaging. 

Continuous ARF pulses can be used to excite the tissue in a steady manner. Selzo et al. [24] proposed a viscoelastic response (VisR) ultrasound imaging method to expedite data acquisition by using ARF impulses. It uses one or two successive ARF impulses to approximate a creep response and to estimate the material relaxation time constant, i.e., the ratio of viscosity to elasticity, τ, of tissue. The VisR method fits the displacements induced by ARF impulses to a mass-spring-damper (MSD) model to estimate τ. This method has the advantages of fast frame rates and independence from tissue inhomogeneity surrounding the ARF region of excitation. However, the acoustic displacement is measured shortly after the ARF excitation by ultrasound, which induces errors to viscosity estimation [25]. The team then proposed methods for error correction and calculation of the elasticity and viscosity relative to the applied ARF [26]. Inspired by these studies, the VisR method may be combined with OCT for high-resolution real-time viscosity measurement of soft tissue. Because the OCT imaging beam can be aligned coaxially with the ARF transducer, the displacement can be measured simultaneously with the excitation. Thus, measurement delay induced error in VisR ultrasound imaging is not an issue when OCT imaging is used.

In this paper, we demonstrate a VisR-OCE method for real-time, non-destructive, in vivo possible viscoelasticity measurement of soft tissue. Using two sequential ARF actuation and displacement quantification by OCT phase analysis, the relaxation time constant is measured almost in real time. The feasibility of the method is evaluated by virtual experiment based on numerical simulation. A tissue mimicking phantom is measured and compared with uniaxial tensile test to evaluate the accuracy of the method. This simple and easy to use VisR-OCE method may provide an effective tool for viscosity characterization of soft tissues in real time and in vivo.

## 2. Materials and Methods

### 2.1. VisR-OCE Measurement System

The VisR-OCE system includes an OCT imaging module and an ultrasonic focusing transducer (1Pφ20F40, Daobo, Changzhou, China) powered by two function generators (DG4202, RIGOL, Suzhou, China) and a voltage amplifier (ATA-2021H, Aigtek, Xi’an, China) as shown in Figure 1a. The ultrasonic transducer is fixed on a linear translation stage so that the height of the transducer can be adjusted precisely to focus on the desired position for ARF loading. The first function generator is triggered by OCT B-scan trigger signal, thus, the time delay between ARF excitation and OCT imaging can be precisely controlled. The first function generator sends out a square wave to control the duration and frequency of ARF impulses. The second function generator is triggered by the rising edge of the output voltage from the first function generator and generates high frequency sinusoidal signals of 1.0 MHz, that is, the central frequency of the ultrasonic transducer. The modulated signals are then amplified by 30 dB by the amplifier and sent to the transducer. The swept source OCT system has a swept frequency of 100 KHz and an axial resolution of ~9 μm in air. Details of the OCT system can be found in our previous publication [27]. A photo of the transducer and OCT scanning lens setup is shown in Figure 1b. An example of the time sequence of the excitation signal is shown in Figure 1c, where two impulses are shown. The frequency and duration of the modulated signal can be adjusted based on the purpose of experiment.

### 2.2. Relaxation Time Constant Measurement

The modulated ARF can be described as a step function multiplied by a series of Dirac functions as:(1)$$F\left(t\right)=A_{d}u\left(t\right)\left(\sum_{n=0}^{\infty }\delta \left(t-n\Delta t\right)\right),$$
where $F\left(t\right)$ is the time function of the ARF, $A_{d}$ is the amplitude of the ARF, $u\left(t\right)$ is a unit step function, Δt is the time between pulses, which is 1 μs determined by the central frequency of the transducer. Because the duration of the ultrasonic pulses is much shorter than the typical viscoelastic soft tissue response, the stress relaxation of the soft tissue during the high frequency ultrasound pulses can be ignored. Thus Equation (1) can be simplified as:(2)$$F\left(t\right)=Au\left(t\right),$$
where *A* is the amplitude equivalent to $A_{d}$ in Equation (1). Thus, the ARF excitation can be taken as a constant force sequence as shown in Figure 2a. The Kelvin-Voigt model consisting of a spring and a damper in parallel, represented by the differential equation of [28] is:(3)$$F\left(t\right)=Ex\left(t\right)+E\tau \frac{dx\left(t\right)}{dt},$$
where x(t) is the displacement, *E* is the Young’s modulus, τ is the relaxation time constant and τ=η/E where η is the viscosity coefficient.

With the initial condition of *x*(0) = 0, the solution *x*(*t*) is:(4)$$x\left(t\right)=\frac{F\left(t\right)}{E}\left(1-e^{-\frac{t}{\tau }}\right),$$

Assume the sample is loaded twice with both the force duration and interval of $t_{0}$ then the displacements are:(5)$$\left\{\begin{array}{c} x\left(t_{0}\right)=\frac{F\left(t\right)}{E}\left(1-e^{-\frac{t_{0}}{\tau }}\right)\\ x\left(t_{1}\right)=\frac{F\left(t\right)}{E}\left(e^{-\frac{t_{0}}{\tau }}-e^{-\frac{2t_{0}}{\tau }}\right)\\ x\left(t_{2}\right)=\frac{F\left(t\right)}{E}\left(1+e^{-\frac{2t_{0}}{\tau }}-e^{-\frac{t_{0}}{\tau }}-e^{-\frac{3t_{0}}{\tau }}\right) \end{array}\right.,$$
where $t_{0}$ is the end of first loading; $t_{1}=2t_{0}$ is the end of the first unloading; $t_{2}=3t_{0}$ is the end of the second loading. In other words, $t_{1}$ is the midpoint of $t_{0}$ and $t_{2}$. A diagram of the loading and unloading process is shown in Figure 2b, where the solid lines from 0 to $t_{0}$ and from $t_{1}$ to $t_{2}$ are the loading displacement curves, and the other two solid lines are the unloading displacement curves. The dashed lines demonstrate the trend of the corresponding curves if the loading or unloading procedure were not interrupted. From Equation (5), it can be obtained that:(6)$$x\left(t_{2}\right)-x\left(t_{0}\right)=x\left(t_{1}\right)\cdot e^{-t_{o}/\tau }.$$

By Equation (6), the relaxation time constant τ is solved as:(7)$$\tau =-ln{\left(\frac{x\left(t_{2}\right)-x\left(t_{0}\right)}{x\left(t_{1}\right)}\right)}^{-1}\cdot t_{0}.$$

Thus, τ is obtained simply by two ARF excitation without complex data fitting or inverse computation.

To measure the displacements in Equation (7), the OCT interference signal is processed by phase analysis method. The effective optical signal $D_{n}$ received by the reflector at depth *z*, when a sample is excited for the *n*th time is:(8)$$D_{n}=R\;cos\left(2k\left(z+\Delta z_{n}\right)+\varphi \right),$$
where *R* is the intensity of the interference signal, *k* is the wave number of the light source, $\Delta z_{n}$ is the displacement at depth *z*, φ is the initial phase of the interference signal. Based on frequency shift principle, the Fourier transform of $D_{n}$ is:(9)$${\hat{D}}_{n}=R\left[\delta \left(z+2z\right)+\delta \left(z-2z\right)\right]e^{j\left(k\Delta z_{n}+\varphi \right)}.$$

The phase shift between the nth A-scan and the first A-scan can be calculated from the cross correlation of $D_{n}$ and $D_{1}$ by:(10)$$\Delta \varphi \left(z,t\right)=angle\left({\hat{D}}^{*}{}_{n}\cdot {\hat{D}}_{1}\right).$$

Then, the displacement at the depth *z* at time *t*, $u_{z}\left(z,t\right)$, is calculated by:(11)$$u_{z}\left(z,t\right)=\frac{\Delta \varphi \left(z,t\right)}{4\pi n}\lambda ,$$
where λ=1.31 μm is the central wavelength of the OCT system, n=1.5 is the refractive index of the specimen. 

### 2.3. Virtual Experiment by Numerical Modeling

Virtual experiment by numerical modeling was conducted to study whether the material properties and experimental parameters influence the measurement accuracy. The modeling includes two steps: ultrasound simulation using Field II simulation program (https://field-ii.dk/) [29] and response of the sample to the ARF using COMSOL (COMSOL Multiphysics v. 5.5, Stockholm, Sweden). The parameters of the transducer including the focal length of 40 mm, the radius of the transducer element of 20 mm and the frequency of 1 MHz were input for simulation. Once the intensity of the acoustic field $\vec{I}$ is obtained as shown in Figure 3a, the acoustic radiation force is calculated as:(12)$$F\left(t\right)=\frac{2\alpha \vec{I}\left(t\right)}{c},$$
where α is the absorption index of the specimen, *c* is the speed of the sound, α  is set as 0.5 dB/cm/MHz and *c* is set as 1540 m/s. Then, the ARF is sampled for point forces and added to the mesh of the finite element method (FEM) model as shown in Figure 3b through LiveLink for Matlab (Matlab 2019b, MA, USA).

During finite element modeling, a 2D model is created with the size of 40 mm × 50 mm as shown in Figure 3b. The model was automatically mashed with 8624 quadrilateral elements with a lowest quality of 0.41 and an average quality of 0.86 by COMSOL (COMSOL Multiphysics v. 5.5, Stockholm, Sweden). Twice and three times the number of the automatic mesh were conducted to evaluate the mesh independence. Similar results were obtained; thus, the automatic mesh was applied in the simulation in COMSOL (COMSOL Multiphysics v. 5.5, Stockholm, Sweden). When the transducer is set at the bottom and focused at the top surface of the specimen, the perfectly matched layer (PML) boundaries are set at the left and right side of the specimen. Free boundary and fixed boundary are set for the top and the bottom surfaces of the specimen. The density of the material is set as *ρ* = 1055 kg/m^2^. The material parameters of the Kelvin-Voigt model, with the Young’s modulus in the range of skin [30] are listed in Table 1. Simulations of the material with the relaxation time constant τ changed from 0.1 ms to 2 ms with a step of 0.1, and the Poisson’s ratio *ν* changed from 0.3 to 0.48 with a step of 0.02 carried out. 

In COMSOL (COMSOL Multiphysics v. 5.5, Stockholm, Sweden), the Domain ODEs (ordinary differential equations) and DAEs (differential-algebraic equations) interface was used to solve the differential equation of the Voigt model [31]. The time and space varying displacements in the vertical direction are then extracted from the simulation results. To testify the OCE method, white Gaussian noise is added to the displacements with a signal to noise ratio of 50 dB. Then the noise corrupted displacement data are input to the OCT imaging model expressed by Equations (9) and (10). After the phase change is calculated, the displacements can be obtained by Equation (11), from which the relaxation time constant *τ* can also be calculated. The displacements obtained by FEM and virtual OCE measurement were compared to evaluate the feasibility of the OCE algorithm. In addition, viscoelastic materials with varying elasticities, viscosities and Poisson’s ratio were simulated to evaluate their influence on the performance of VisR-OCE method. A flow chart of the process of the virtual experiment is shown in Figure 4.

### 2.4. Soft Tissue Phantom Fabrication and Relaxation Time Constant Measurement

Soft tissue phantoms were made by mixing AB liquid silica gel (184, Dow Corning, Pasadena, TX, USA) and 0.1% TiO_2_ particles (P25, 20 nm, Macklin, Shanghai, China) as optical scatter in a tube. The tube was put in an ultrasonic cleaning machine (KQ218, Kunshan Ultrasonic Cleaning Co., Kunshan, China) for 30 min to fully mix the materials. Then, the mixture was poured into a container and kept in a vacuum box for an hour to remove the air bubbles and solidify. During measurement, the specimen was fixed between the transducer and a flat plate with a hole in the center. The axes of the OCT scanning lens and the ARF transducer were aligned and focused at the same spot at 1 mm below the top surface of the specimen. Based on the purposes of experiment, single or double ARF impulses with a certain duration were applied, OCT interference signal was acquired simultaneously or with a deliberate time difference. Each experiment was repeated 3 times. To validate the proposed VisR-OCE method for relaxation time constant measurement, uniaxial tensile test of a piece of the specimen was conducted. An in situ symmetrical tensile tester (IBTC-100, CARE Measurement & Control, Tianjin, China) was employed to measure the stress-strain, stress-time and strain-time curves. By nonlinear curve fitting of the integral constitutive model of
(13)$$\varepsilon \left(t\right)=\int_{0}^{t}\frac{1}{E}\left(1-e^{-\frac{E}{\eta }\;\left(t-\tau \right)}\right)\frac{d\sigma \left(\tau \right)}{dt}d\tau ,$$
where *E* is the Young’s modulus and *η* is the viscosity coefficient, the Young’s modulus and relaxation time constant can be obtained.

## 3. Results

### 3.1. Virtual Experiment by Numerical Modeling

Time varying axial displacements of two sets of points are extracted. The points are distributed evenly in the horizontal and vertical directions, respectively. Both directions pass through the focal spot. A total of 20 horizontally distributed points across a distance of 2 mm starting from the focal spot and 200 vertically distributed points across a distance of 2 mm centered by the focal spot are examined. The 3D map of the displacement is shown in Figure 5a. It can be seen that the peak of the displacements appears at the starts of the second unloading, which is ~3 μm. 

The displacement distribution was then added by 50 dB Gaussian white noise and input to the OCT model according to Equation (8). The phase map was obtained as shown in Figure 5b. Because the maximum imaging depth is ~2 mm, only the phase distribution of the top 2 mm was calculated. The phase distribution along a horizontal line at 1 mm depth is plotted in the lower figure in Figure 5b. After phase unwrapping, the displacements were calculated according to Equation (11) and plotted in Figure 5c. It can be seen that the original applied displacements match well with the one obtained by OCE. It demonstrated the robustness of the OCE algorithm to noise.

Simulation was also performed with varying viscosities. The displacement curves are shown in Figure 6a. It can be seen that the higher the *τ*, the less the displacement restores after unloading. To investigate the relationship between the displacements and the loads, a parameter of displacement increase rate, *k*, was defined as:(14)$$k=\left(x\left(t_{3}\right)-x\left(t_{2}\right)\right)/x\left(t_{1}\right)$$
where $\;x\left(t_{n}\right)$ is the displacement at the *n*th loading or unloading time. Changes of *k* versus *τ* are plotted in Figure 6b, which shows that the low viscosity material can almost fully recover to the initial stage before the second loading. For high viscosity material, the displacement recovers slowly, thus, $x\left(t_{2}\right)$ is high, and *k* is low. The duration of the load, $t_{0}$, also influences the displacements. As shown in Figure 6b, when the duration is 1 ms, the difference between the high and low viscosity is largest. The duration of 1 ms can be helpful for differentiating the materials with different levels of viscosity. Under different loading durations, the relative errors changing with *τ* are plotted in Figure 6c. The relative error is calculated by $relative\;error=\frac{\tau_{OCE}-\tau }{\tau }\times 100\%$, where the $\tau_{OCE}$ is the value obtained by virtual OCE experiment described in Section 2.3, and *τ* is the value given for simulation. From Figure 6c, it can be seen that 0.25 ms for low viscosity material, and 1 ms for high viscosity material are good choices for experiment in terms of error reduction.

To evaluate the influence of the elasticity, Poisson’s ratio, *ν*, and viscosity to the measurement error of the VisR-OCE, two specimens with the Young’s modulus of 13 kPa and 130 kPa loaded by 1 ms duration of ARF impulses are simulated. Figure 7 shows the viscosity measurement error changes with *ν* and *τ* for the two specimens. Figure 7a shows that the average error of low viscosity (*τ* = 0.50 ms) and high viscosity (*τ* = 2.00 ms) are 1.62% and 2.64%, respectively, with the Poisson’s ratio changing from 0.3 to 0.48. Figure 7b shows that average errors are 1.60% and 3.80%, respectively, which is similar to Figure 7a. It indicates that the proposed method is not sensitive to the Poisson’s ratio and the elasticity of the material.

### 3.2. Soft Tissue Phantom Measurement

Two phantoms with different viscosity were made following the procedure described in Section 2.4. A photo of one of them is shown in Figure 8a. A cross-sectional (B-mode) OCT image is shown in Figure 8b and an M-mode image along the dashed line in Figure 8b is shown in Figure 8c. Figure 8b,c shows the structure of the specimen. To test if the specimens display viscoelastic properties, a series of experiments were conducted with single ARF impulse excitation while the duration of the impulses changed from 0.5 ms to 5 ms with a step of 0.5 ms. During measurement, ARF excitation was set 10 ms behind the starting of OCT imaging so that reference signals of the specimen at non-loading condition were recorded at the first 10 ms. The OCT acquisition rate is 100 A-lines/ms. The OCT displacement maps under various excitation durations are shown in Figure 9a, where the subfigures from A to J are excitation durations from 0.5 ms to 5 ms with a step of 0.5 ms. It can be seen that the displacements along the depth has similar pattern, which indicates that the ARF focal region is elliptical with the longer axis in the vertical direction, and the acoustic field is fairly constant. Figure 9b shows that the time of sample displacement rise is highly consistent with the actual loading time. With the increase of the duration time, the peak displacement increases, which is consistent with the dynamic behavior of viscoelastic materials.

To measure *τ* of the specimen, three excitations with duration of 1 ms were applied, where only the first two excitations were used for relaxation time constant, *τ*, calculation, and the 3rd one is for observation. The OCT phase map and the displacement curve versus time are shown in Figure 10. The peaks of the displacement at the corresponding moment were obtained, which was input to Equation (5) to calculate *τ*. Each experiment was repeated three times, and the *τ* solved is 7.9 ms ± 0.005 ms for the 1st specimen and 6.3 ms ± 0.005 ms for the 2nd specimen.

To evaluate the accuracy of the results obtained by the proposed method, another specimen made by the same material as the first specimen was also measured by uniaxial tensile test. Results are shown in Figure 11. It can be seen that the stress is almost linearly related to the strain under a constant strain rate. The integral constitutive model expressed by Equation (13) can describe this relationship well. By curve fitting, τ of the first specimen is calculated as 7.5 ms. The discrepancy between the VisR-OCE and the tensile measurement results is ~3%.

### 3.3. Pork Meat Measurement

A pork specimen with 5 cm × 5 cm × 3 cm in length, width and height was measured at room temperature. The specimen was cut from a piece of pork meat bought from a supermarket in China. The experiment was conducted right after the meat was bought, without any preserving time. A similar procedure to the phantom measurement was carried out. It took less than 1 min to complete the test, thus no hydration was applied. A setup of the sample with the ultrasound transducer fixed underneath it and the OCT scanning lens aligned on the top is shown in Figure 12a. Prior to the test, OCT B-mode imaging was carried out to find a homogeneous region for viscosity measurement. The sample was also loaded by three ARF impulses and the displacement distribution is shown in Figure 12b. From the top figure, it can be seen that reasonable data are obtained within the 1 mm depth near the surface, due to the penetration limitation of OCT imaging. From the displacements induced by the ARF impulses, the τ of the pork meat is obtained. The experiment was repeated at three different regions. The average value of τ was measured to be 6.2 ± 0.03 ms, which is in the range of pork meat measured by nuclear magnetic resonance technique [32].

## 4. Discussion and Conclusions

We have shown that the VisR-OCE based on one or two ARF impulses is effective for relaxation time constant measurement in a non-destructive manner and possibly in real time. Compared to the traditional methods, obtaining high resolution and real time measurements allows for quick detection of localized changes of soft tissue. It will not only facilitate in vivo imaging when keeping still for a long time is not preferred but will also be useful in evaluating the effectiveness of some treatment methods. 

The virtual experiment based on numerical simulation provides not only an assessment for the feasibility of the VisR-OCE algorithm but also a solution for optimal experimental parameters determination. The soft tissue is assumed homogenous locally in this study. Thus, a 2D simulation of a sector of the region centered by the impact axis can well reflect the response of the whole specimen. When heterogenous material is measured, 3D simulation may be required considering the loading factors, such as the diameter of the ARF focal spot. Localization of the focal spot of the ARF is one of the key factors for accurate measurement results. Through the echo function of the pulse generator, the focus can be located by constantly shifting a plate until the strongest echo appears. The variation of the sound speed in different media should be considered when measuring different specimens for accurate localization of the focus. Figure 5 shows that the displacements input to the OCT model matches well with the displacement measured based on OCT phase analysis through the depth of 2 mm. Therefore, the proposed method has the potential to measure layered structures within 2 mm thickness. Due to the decrease of the signal to noise ratio at the locations farther away from the focal spot, data from the focal spot is preferred for the calculation of the relaxation time constant. It can be seen from Figure 7 that, with the increase of viscosity, the duration of loadings should also increase in order to reduce measurement error. The results are not influenced much by the Poisson’s ratio and the elasticity of the material, which indicates that the proposed method is possibly suitable for a wide range of soft materials.

The pork specimen measurement shown in Figure 12 demonstrated the feasibility of the VisR-OCE method for viscosity characterization of biological tissue. Only one sample was measurement, and the postmortem interval of the pig was not known. More rigorous research will be planned to study the biomechanical properties of soft tissue. Another limitation of the current study is that only the relaxation time constant is obtained. The elastic modulus and viscosity of the Kelvin-Voigt model may be quantified separately by including additional experimental and analytical steps. Studies have shown that other rheological models may be superior to the Kelvin-Voigt model for a wide frequency range. If and how the VisR-OCE can be developed to measure the parameters of other rheological models will be studied in the future [33,34]. 

In conclusion, a fast and high-resolution viscosity measurement method, VisR-OCE, is proposed. The measurement can be completed in several milliseconds with remote excitation of ARF and OCT imaging. It provides a possible solution for non-destructive and fast viscosity test of biological tissues. Based on the proposed numerical simulation method, the optimal duration for ARF radiation is 0.25 ms and 1 ms for relatively low and high viscous specimens, respectively. The error for displacement measurement is less than 5%, and the error of relaxation time constant measurement for soft tissue is ~3% compared with tensile test. This measurement method may provide an effective tool for viscosity quantifications in biomedicine, biomaterials design and fabrication.

## Figures and Tables

**Figure 1 materials-16-06019-f001:**
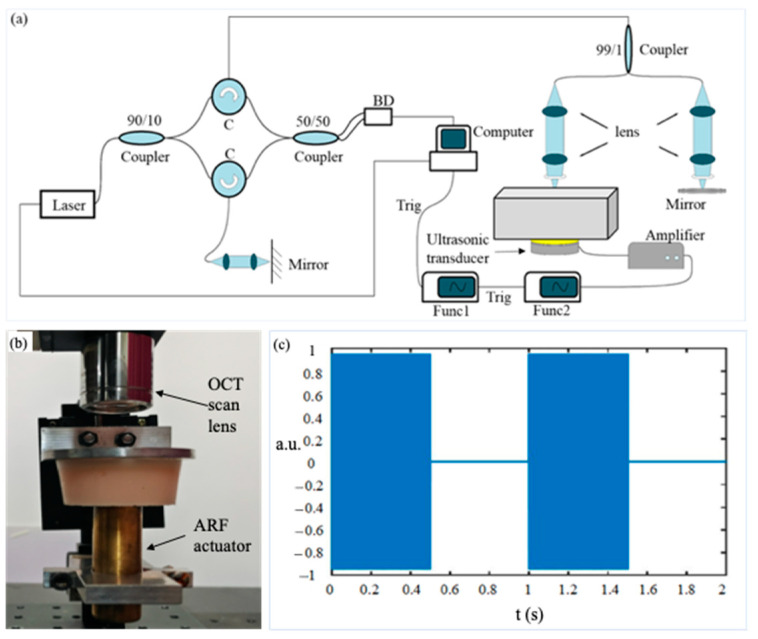
Experimental setup. (**a**) A schematic diagram of the VisR-OCE system. The meanings of some of the letters are as follows: C: circulator, BD: balanced photodetector, Trig: trigger signal, Func1: the 1st function generator, Func2: the 2nd function generator; (**b**) a photo of the ultrasonic transducer and the OCT imaging lens; (**c**) a diagram of the time sequence of the ARF impulses.

**Figure 2 materials-16-06019-f002:**
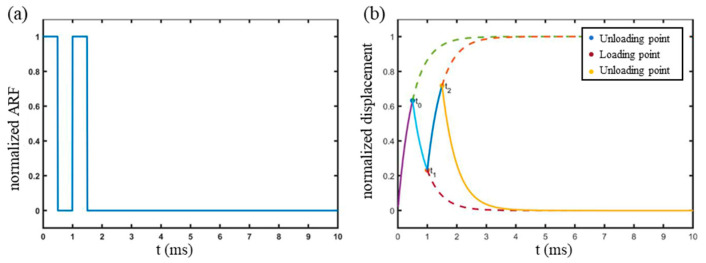
Schematic diagram of the double ARF impulse loading process. (**a**) Normalized ARF impulses with loading and unloading durations of 0.5 ms for each impulse. (**b**) Normalized displacement response of a soft tissue sample.

**Figure 3 materials-16-06019-f003:**
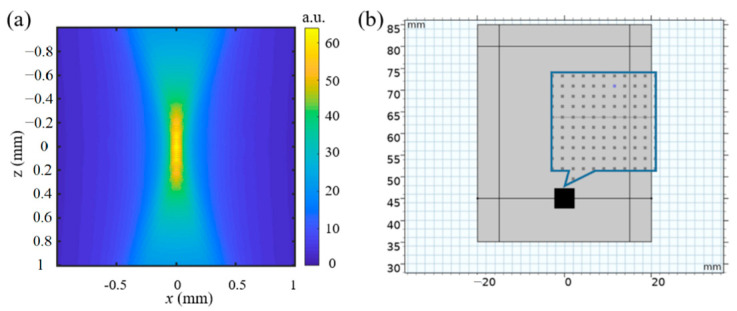
(**a**) Sound pressure distribution in the focus range of 1 mm × 1 mm; (**b**) FEM model with the ARF applied.

**Figure 4 materials-16-06019-f004:**
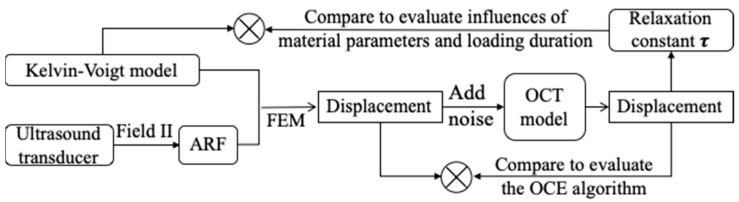
A flow chart of the virtual experiment for evaluation of the OCE algorithm and influences of the material parameters and loading duration to the measurement accuracy.

**Figure 5 materials-16-06019-f005:**
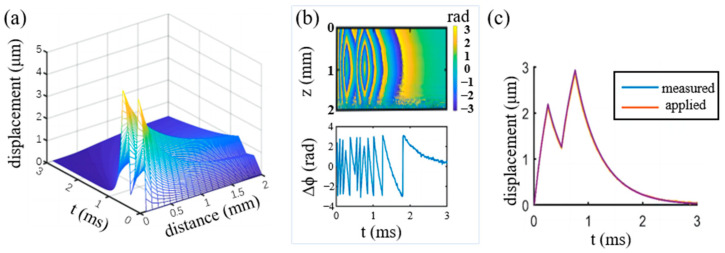
Finite element modeling of ARF excitation and displacement measurement by OCE. (**a**) Spatial and temporal distribution of the displacement distribution in the excitation region; (**b**) the phase shift distribution caused by the displacements in (**a**) according to OCE (upper figure), and the phase distribution at 1 mm depth (the lower figure); (**c**) comparison of the displacement curves of the applied and measured by OCE virtually.

**Figure 6 materials-16-06019-f006:**
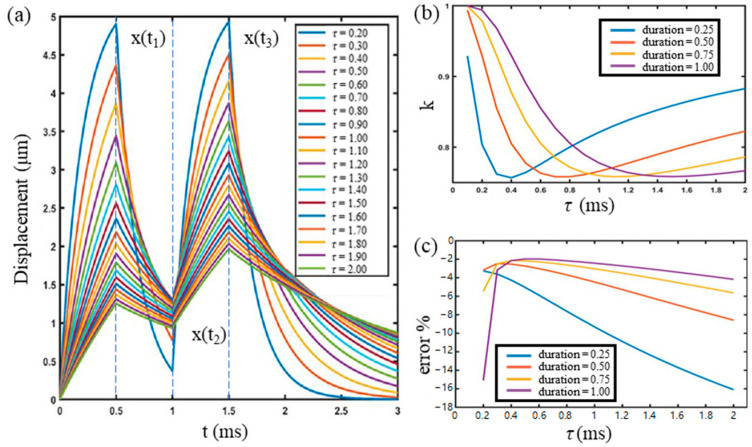
(**a**) Curves of displacement changing with time for specimens with different viscosities, when the duration of each ARF loading impulse is 0.5 ms. The unit of *τ* on the figure is ms; (**b**) curve of the increase rate *k* versus τ for different loading duration; (**c**) relative measurement error of τ for different loading duration.

**Figure 7 materials-16-06019-f007:**
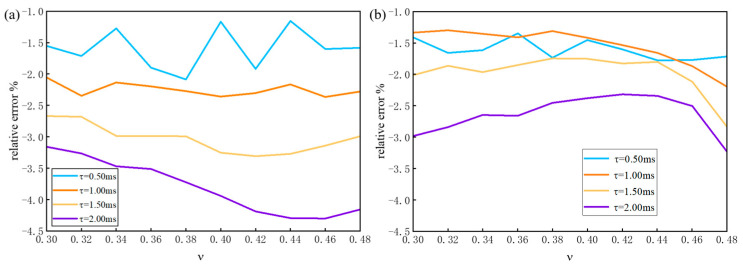
Changes of the relative measurement error versus the Poisson’s ratio and relaxation time constant, τ. (**a**) Error curves of material with elasticity E=13 KPa; (**b**) error curves of material with elasticity E=130 KPa.

**Figure 8 materials-16-06019-f008:**
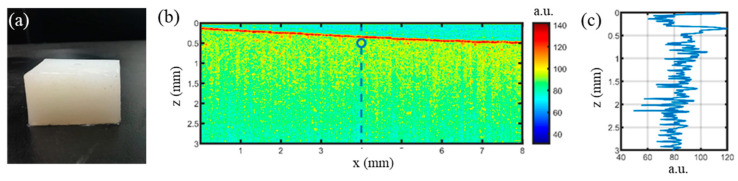
OCT imaging of a phantom. (**a**) A photo of one of the phantoms. (**b**) An OCT B-mode image of the specimen; (**c**) an OCT M-Mode image along the dashed line in (**b**).

**Figure 9 materials-16-06019-f009:**
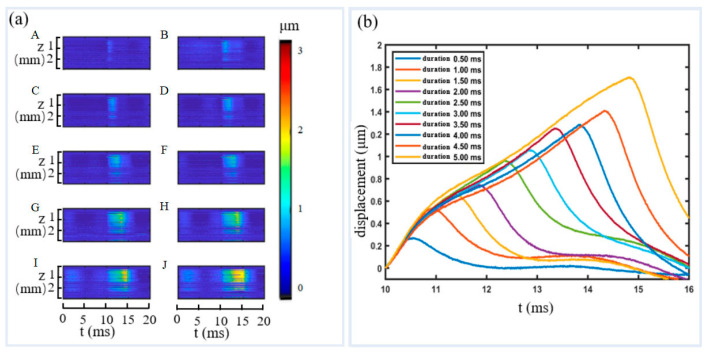
Displacements generated by different duration of ARF and measured by OCT. (**a**) OCT maps of the specimen under different duration of excitation. (**b**) Changes of displacements at the focal point versus time.

**Figure 10 materials-16-06019-f010:**
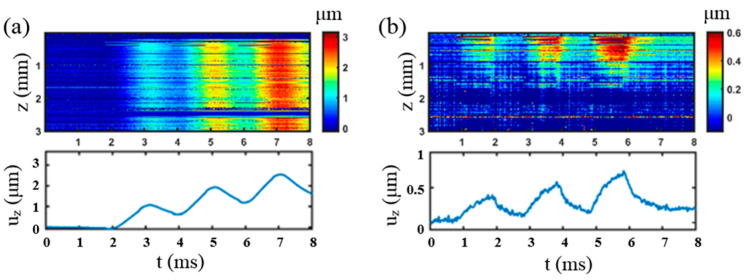
Viscosity measurement of two phantoms. (**a**) Displacement distribution of the 1st phantom. (**b**) Displacement distribution of the 2nd phantom. In both (**a**,**b**), the z on the top figure represents depth. The bottom figures are the displacement plots at one certain depth, that is, along one horizontal line of the top figure.

**Figure 11 materials-16-06019-f011:**
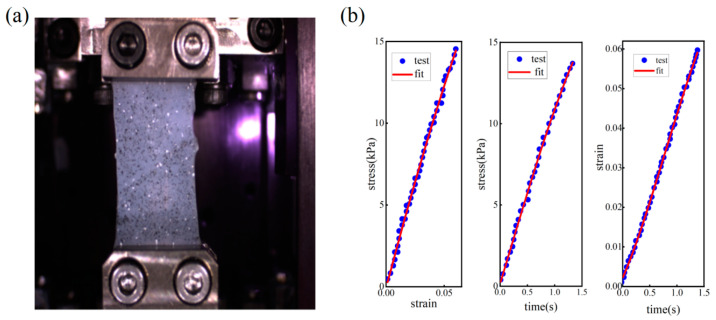
Uniaxial tensile test. (**a**) A photo of the specimen under tension. (**b**) The curves of stress-strain, stress-time, strain-time obtained and the curve fitting results.

**Figure 12 materials-16-06019-f012:**
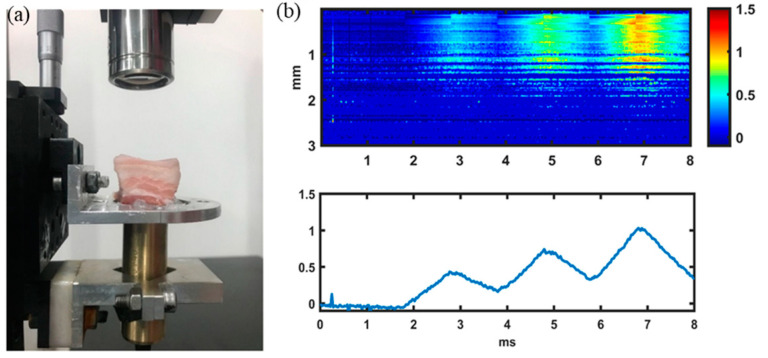
Viscosity measurement of a pork meat sample. (**a**) A photo showing the experimental setup; (**b**) displacement measured, where the top figure is the displacement distribution within 3 mm of depth while imaged for 8 ms. The bottom figure is the displacement plot vs. time at a certain depth.

**Table 1 materials-16-06019-t001:** Viscoelastic parameters for simulation.

Voigt Model	E (kPa)	τ (ms)	ν
	13	Range (0.1, 0.1, 2)	Range (0.3, 0.02, 0.48)
	130	Range (0.1, 0.1, 2)	Range (0.3, 0.02, 0.48)

## Data Availability

Data presented in this paper are available upon reasonable request.

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
