# Peer review of "Real-Time Nondestructive Viscosity Measurement of Soft Tissue Based on Viscoelastic Response Optical Coherence Elastography"

_materials, 2023, doi:10.3390/ma16176019_

Round 1

Reviewer 1 Report

This paper deals with a method for measuring the viscosity of soft tissues. It is based on the viscoelastic response induced by the acoustic radiation force and optical coherence tomography imaging. Generally speaking, the article is well written and fits with the scope of the journal, so it could be considered for publication after submitting the revised following the issues listed below.

Authors should carefully review the English language. Some grammatical errors were found throughout the manuscript.

Authors mention in the whole text the term “viscosity”; however, such a property is never reported using the corresponding units (i.e., cSt, Pa.s, etc.,)

Latin terms such as “in vivo”, “in situ”, etc. should be written in italics.

P2, lines 94-95: Usually, experimentation is used for validating the numerical findings.

P4. Line 137: please check nomenclature (tao or tao sub sigma?)

P5, line 165: More details about the numerical simulation is needed. If the finite element method is used, information about meshing, mesh independence, etc. should be provided.

P5, line 182: What was the reason for considering a 2D analysis?

P6, Table 1: Where do this data come from? Please provide a reference.

P6. Line 207: “Dow Corning” instead of “Dowcoring”

P10, line 328: Section 3.3 is incomplete. Results obtained with the developed method should be compared with data reported elsewhere..

P10, line 328: A piece of pork meat was used as sample (not a pork).

P10-11, Discussion and conclusions: What is the advantage of obtaining real time measurements compared to the traditional methods reported in the literature?

P11-13: References should be listed using a uniform format. Some sources titles are abbreviated and other are fully written.

Some minor grammar mistakes were found throughout the manuscript.

Reviewer 2 Report

In the Manuscript “Real-time nondestructive viscosity measurement of soft tissue 2 based on viscoelastic response optical coherence elastography , prepared by Zhixin Liu, a real-time and high-resolution viscosity  measurement method based on viscoelastic response induced by acoustic radiation force (ARF) and  optical coherence tomography (OCT) imaging is presented. The proposed method may offer an effective tool for fast nondestructive viscosity test of biological tissues.

Comprehensive state-of the art regarding to modeling of visco-elastic of biological tissues is presented.

Manuscript may be acceptable for publishing after minor revision.

The following issues should be resolved:  

1.      All the materials used in this investigation have to be listed in Materials and Methods section, along with their basic properties, supplier, purity, etc.

2.      Sample preparation for should be described in details and given.

3.      The authors should clearly sate which sample have been used for each experiment.

it is ok 

Reviewer 3 Report

This article summarizes an interesting study with potentially valuable findings. However, it is unacceptable for publication in its current form because of a number of false or misleading statements, mainly in the introduction, and a lack of sufficient detail in some places to describe the methods used.

There are some debatable assertions or opinions put in the Introduction, as if they are fact, which are not substantiated. Such opinions do not belong in a technical article. For example, from line 65-66: “There are too many factors affect the measurement results to make this kind of method robust for viscosity measurement.” This is an opinion, not a fact.

Also, it is stated that “almost all the material measured were assumed as Kelvin-Voigt viscoelastic model..”, and that, essentially, this is the best rheological model from soft tissue. This is false. Many studies have shown that the Kelvin-Voigt model is inferior to other rheological models. For example, see the following articles and some of the references they provide, many which have shown the value of rheological models based on linear fractional derivatives.

1)    Yasar TK, Royston TJ, Magin RL. Wideband MR elastography for viscoelasticity model identification. Mag. Res. Med. 70, 479 – 489 doi: 10.1002/mrm.24495 (2013).

2)    Kearney SP, Khan AA, Dai Z, Royston TJ. Dynamic viscoelastic models of human skin using optical elastography. Phys. Med. Biol. 60, 6975-6990 (2015).

In Section 3.2 detail should be provided about how the soft tissue phantoms were made. What material? What fabrication process? Have there been other studies in the literature on this particular phantom material? It is important that someone should be able to repeat the presented study.

In Section 3.3 detail on the timing of the study on the pork specimen should be provided. How was it handled? What temperature? Any preservative treatment? All of these can affect viscoelastic behavior of an excised tissue sample.

The presented experimental studies demonstrate dissipative behavior of the materials studied, but really do not provide the type of information needed to identify and evaluate rheological modeling options, beyond identifying a relaxation time constant based on modeling assumptions.

Round 2

Reviewer 3 Report

I am satisfied with the revision.

Author Response

Thank you for your approval of our work.